# Production of Bioactive Compounds with Broad Spectrum Bactericidal Action, Bio-Film Inhibition and Antilarval Potential by the Secondary Metabolites of the Endophytic Fungus *Cochliobolus* sp. APS1 Isolated from the Indian Medicinal Herb *Andrographis paniculata*

**DOI:** 10.3390/molecules27051459

**Published:** 2022-02-22

**Authors:** Hiran Kanti Santra, Santanu Maity, Debdulal Banerjee

**Affiliations:** Microbiology and Microbial Biotechnology Laboratory, Department of Botany and Forestry, Vidyasagar University, Midnapore 721102, West Bengal, India; hiran.kanti@gmail.com (H.K.S.); rajdeep14041993@gmail.com (S.M.)

**Keywords:** antibacterial, aziridine, 1-(2-aminoethyl)-, optimization, anti-biofilm, larvicidal potency

## Abstract

Endophytes, being the co-evolution partners of green host plants, are factories of pharmaceutically valuable novel natural products. *Cochliobolus* sp. APS1, an endophyte of *Andrographis paniculata* (Green Chiretta), produces a plethora of natural bioactive compounds and the multipotent alkaloid Aziridine, 1-(2-aminoethyl)-, is the prime one among them. The isolate exhibited antibacterial, anti-biofilm, and antilarval potency. The MIC and MBC values of the ethyl-acetate culture extract ranged from 15.62 to 250 µg/mL against ten pathogenic microorganisms (including MRSA and VRSA). Killing kinetics data along with the leakage of macromolecules into the extracellular environment supports the cidal activity of the antibacterial principles. The broad spectrum antibacterial activity of Aziridine, 1-(2-aminoethyl)-, was optimized by a one-variable-at-a-time system coupled with response surface methodology, which led to a 45% enhancement of the antibacterial activity. The maximum response (22.81 ± 0.16 mm of zone of inhibition against MRSA) was marked in 250 mL Erlenmeyer flask containing 90 mL potato dextrose broth supplemented with (g%/L) glucose, 9.7; urea concentration, 0.74; with medium pH 6.48; after 8.76 days of incubation at 26 °C. APS1 strongly inhibited biofilm formation in the tested pathogenic microorganisms and acts as a larvicidal agent against the Dengue-vector *Aedes aegypti*. This is probably the first report of Aziridine, 1-(2-aminoethyl)-, from any endophytic source. *Cochliobolus* sp. APS1 possesses industrial importance for the production of bioactive alkaloids.

## 1. Introduction

Endophytes are the silent microbial partners of the plant, residing within the host tissues for a part or the entirety of their life cycle, and help the plant to withstand biotic or abiotic stresses. The host and its endophytes are always engaged in a symbiotic relationship where plants provide nutrients to the microbes and the microbial flora assists their eukaryotic counterpart by acting as a line of defense against parasitic pathogens, either directly by enhancing host plants’ resistance or by being engaged in chemical warfare [1,2,3]. Plants and their endophytes are co-evolutionary partners and thus share valuable genes. They are factories of novel bioactive compounds with unique biotechnological importance in the control, prevention and amelioration of fatal diseases by producing antimicrobial, antioxidative, and anticancer compound sand industrially valuable enzymes [4,5,6,7,8]. Hydrocarbon derivatives from endophytic sources with bio-fuel potency are popularly known as myco-diesel and provide a glimpse into the ecology of fungus–plant and environmental relationship [9].

In the modern world, antimicrobial compounds are one of the most used chemotherapeutic agents, but they are losing their superiority due to the development of antimicrobial resistance. Antimicrobial resistance (AMR) towards existing drugs is a matter of serious concern, especially regarding infections caused by MRSA (methicillin-resistant *Staphylococcus aureus*), VRSA (vancomycin-resistant *Staphylococcus aureus*), PRSA (penicillin-resistant *Staphylococcus aureus*) and other MDR (multi-drug resistant) strains, which are worsening the situation and the failure of pre-existing molecules emphasizes the need to search in untapped fields. The hunt for safer and unique drugs (novel natural products) of plant or microbe origin from untapped biological sources is the best probable way out to meet the need of pharmaceutical sectors to combat serious health issues [10].Endophyte biology is opening up a new domain of solutions to the pharmaceutical world and offers a variety of bioactive compounds.

A much more severe pathogenicity and failure of antimicrobial agents is reported from strains forming biofilms in medical devices (prosthetic device and catheter) causing genitourinary tract and skin osteoarticular infections, endocarditis, pneumonia, and toxic shock syndrome spread from nosocomial or hospital-aided infections [11]. Type 1 fimbriae, pili, flagella and bacterial toxins mediate the adherence of planktonic cells forming a matrix type coating to induce biofilm formation, making it resistant to the host defense system and also restricts the penetrance of immunological factors to the bacterial cells. Biofilms promote the increased survival of pathogenic cells on the gut epithelial surface, prostheses, internal and external body organs, catheters or other intravenous and implanted devices, etc. [12]. Natural agents are highly desirable to tackle these situations. Essential oils and other bioactive natural components have been proposed to ameliorate this problem. Bio-film inhibition is a serious mode of dealing with microbial infection as it enhances the drug resistance phenomenon of bacterial colonies enabling quorum-sensing strategies. 

Mosquitoes, as the most potent vector of several fatal diseases, such as dengue, chikungunya, malaria, filaria, etc., are a serious threat in tropical and subtropical countries, such as India, South Africa, Brazil, etc. Dengue raises a serious issue, infecting almost 3.9 billion people in 128 countries [13]. These vectors can only be restricted by the synthetic chemicals xenobiotics, such as organochlorines, organophosphates, carbamates, DDTs, etc., which possesses severe side effects [14]. Thus, to maintain the sustainability of ecosystem, components with less or zero toxicity to non-target organisms should be adopted. Natural products from fungal or other biological sources are cost effective, environmentally friendly and play a significant role in ameliorating this type of problems [15]. Ragavendran et al. (2019) reported numerous natural bioactive products: pthalic acid, 1-nonadecane, bendazol, carbonic acid and boron trichloride with larvicidal (against fourth instar larvae of *Culex quinquefasciatus* and *Aedes aegypti*) activity from an indigenous soil fungi *Penicillium* sp. [16]. 

The present work reports the antibacterial, anti-larval, and antibiofilm activities of an endophytic isolate of *Andrographis paniculata* (kalmegh) of the Acanthaceae family, a medicinally important Indian herb.The plant *Andrographis paniculata* (family: Acanthaceae) commonly called “Green Chiretta” is selected for isolation of endophytic fungi based on its immense medicinal importance in traditional and ethnic Indian Ayurvedic treatment as an antibacterial and antioxidative agent [17]. The endophytes of kalmegh represent a plethora of natural novel bioactive components widening its industrial and medicinal acceptability. In this study, the endophyte *Cochliobolus* sp. APS1 (isolate code) produced the aziridine alkaloid and 24 other bioactive components that possesses broad spectrum antibacterial action against 10 human pathogenic microorganisms and especially MRSA (methicillin-resistant *Staphylococcus aureus*) and VRSA (vancomycin-resistant *Staphylococcus aureus*). The antibacterial principles were thermostable and non-proteinaceous in nature and possess a bactericidal mode of action. The bioactive components block the necessary enzymes of the central carbohydrate metabolism of the bacterial pathogens and cause membrane leakage, releasing biological macromolecules (DNA and protein) and K^+^ ions. APS1-originated biomolecules also restrict the biofilm formation of bacterial pathogens, possess a larvicidal action towards *Aedes aegypti* third and fourth instar larvae and block the necessary enzyme acetyl-choline esterase. The antibacterial aziridine alkaloid along with other bioactive components (major components from endophytic *Cochliobolus* sp. APS1) are optimized using BBD (Box–Behnken design) and RSM (response surface methodology). An elevation of 45% of the antibacterial action was noticed after the optimization of the fermentation parameter than the pre-optimized one. Our present investigation is unique in terms of its investigation of the anti-bacterial and anti-oxidative action of novel natural products from endophyte origin.

## 2. Results

### 2.1. Identification of the Isolates

The plant was collected from virgin forest patches of Tapobon, Midnapore, West Bengal, which is a store house of ethnomedicinally valuable plants and on their variety the local ethnic tribe depends as the flora of the Tapobon forest is used for as primary treatment against common illnesses. 

In total, five endophytic fungi were isolated from stem (three isolates) and leaf (two isolates) tissues of the explant. No bacterial colonies were observed, as the medium was previously supplemented with the antibiotic streptomycin. Four isolates produced reproductive structures and were identified (seen under light compound microscope) as *Fusarium* sp., *Aspergillus* sp., *Penicillium* sp., and *Rhizopus* sp. (data not shown), but one isolate did not produce any reproductive structure even after growing on a CLA (Carnation leaf pieces agar) medium. Thus, the isolate with no reproductive structure describes its true endophytic nature. CLA is considered as the popular medium that can induce reproductive structure formation. 

The true endophytic nature of the isolate was determined by the tissue finger printing method where the aliquots with wash liquid of the explants were plated on PDA and there was no occurrence of any endophytic fungi. Thus, the non-epiphytic nature of the endophyte was confirmed. As the endophytes were isolated from the healthy (external appearance was disease free) and mature plant parts of the *A*. *paniculata*, it was confirmed that the isolate is a true endophyte and non-pathogenic in nature. 

The isolate with no reproductive structure was subjected to rDNA sequencing and finally it was identified as *Cochliobolus* sp. APS1 (Gen Bank Acc. No.-MH102384.1). The constructed phylogenetic tree provides information about the identity of the endophytic fungi studied for its bioactivity (Figure 1a). Plate morphology (top view) along with scanning electron microscopic images of sterile mycelial structures are represented in Figure 1b,c, respectively. 

### 2.2. Antibacterial Activity of the Endophytic Fungal Isolates

The antibacterial activity of the five endophytic isolates was determined by the agar well diffusion technique against ten pathogenic strains of Gram-positive and Gram-negative bacteria (Appendix A). Out of the five isolates, *Cochliobolus* sp. APS1 exhibited broad spectrum antibacterial activity (in terms of clear zone of inhibition in mm) against all the Gram-positive (MRSA, VRSA, *B*. *cereus*, *B*. *subtilis*, and *S. aureus*) and Gram-negative (*P*. *mirabilis*, *P*. *aeruginosa*, *S. flexneri*, and *E*. *coli*) bacterial pathogens, except *Vibrio parahaemolyticus*. A broad-spectrum antibiotic ciprofloxacin was used as a positive control in this case.

### 2.3. Thermostable and Non-Proteinaceous Nature of the Antibacterial Components

Proteinase K treated and heat killed EA fractions of APS1 showed prominent clear zones of inhibition against the Gram-positive and Gram-negative bacterial pathogens. The antibacterial action (clear zone of inhibition in mm) was similar to the untreated EA fraction of APS1. Thus, it is proved that the bioactive antibacterial components are thermostable in nature and can tolerate heat treatment. They are also non-proteinaceous in nature as they showed similar degree of antibacterial activity even after proteinase K treatment. There were no zones of inhibition in the case of the control set, which contained uninoculated broth and 1 mg/mL of proteinase K, respectively.

### 2.4. Selection of the Appropriate Extraction Agent

Three different fungal culture broths, malt extract (ME) broth, Czapek Dox (CD) broth and potato dextrose (PD) broth, were tested in this study for endophytic fungal growth and antibacterial activity. Finally, PDB, as the highest producer of an antibacterial clear zone of inhibition, was selected for solvent extraction (Appendix A). Four different solvents, ethyl ether, petroleum ether, n-hexane and ethyl acetate, were used for the extraction of the maximum amount of antibacterial components from the fungal culture broth and finally, ethyl acetate (EA) proved to be the best extraction agent in this respect (Appendix A).The EA fraction of APS1 secondary metabolites (in different concentrations) exhibited prominent clear zones of inhibition against nine pathogenic (broad spectrum) Gram-positive and Gram-negative bacteria, except *Vibrio parahaemolyticus*. The antibacterial activity against Gram-negative *V*. *parahaemolyticus* was similar in the case of both the cell free extract and EA fraction. So, ethyl acetate was used further in this study.

### 2.5. MIC and MBC Determination

The EA fraction of APS1 was tested for its antibacterial efficacy by counting the number of bacterial CFUs. The reduction in bacterial CFUs in comparison to control were high in case of Gram-positive bacteria than the Gram-negative bacteria. The antibacterial principles were considered to be bactericidal in nature if the results exhibited >3 log CFU reduction (in comparison to control) and bacteriostatic if exhibited <3 log CFU reduction (in comparison to control). The MIC and MBC values of EA fraction of APS1 ranged 15.62 µg/mL–125 µg/mL and 62.5 µg/mL–250 µg/mL, respectively. A group of dreadful pathogenic bacteria belonging to MDR strains (multi-drug resistant), MRSA (methicillin-resistant *Staphylococcus aureus*) and VRSA (vancomycin-resistant *Staphylococcus aureus*) were found to be inhibited by the ethyl acetate culture extract efficiently with a MIC and MBC value of 62.5 µg/mL and 125 µg/mL, respectively (Table 1).

### 2.6. Effect on Bacterial Growth Kinetics

The changes in CFU number over incubation hour were recorded as a time-kill curve by treating the active bacterial cultures with respective MIC (MIC/2, MIC*2, MIC*4) values (Figure 2). The culture extract (ethyl acetate fraction) affected the growth of the tested bacterial pathogens in a concentration- and time- (duration of exposure to culture extract) dependent manner. A drastic fall on the viable bacterial cell count happened when treated with concentrations four times higher than MIC values. Gram-positive bacterial pathogens were more sensitive to antibacterial principles than the Gram-negative ones. The result proves that endophytic *Cochliobolus* sp. APS1 ethyl acetate culture extract at MIC values and values higher than MIC or MBC ameliorates the bacterial population drastically, even in the case of the two most fatal pathogens, MRSA and VRSA.

### 2.7. Leakage of Intracellular Materials

The broad-spectrum antimicrobial principles produced by APS1 leads to the leakage of intracellular macromolecules, such as DNA, protein and K^+^ ion content that specifies the cidal effect of the active components causing the lysis of the treated cells. The cell bursting effect was more prominent in the case of Gram-positive pathogens than in the Gram-negative ones. The results were satisfying in the case of MRSA and VRSA than in *P*. *mirabilis* and *P*. *aeruginosa*. There was a two-fold increase (*p* < 0.001) of DNA and protein content in the extracellular environment leaked out from the treated bacterial cells after 24 h of treatment in a comparison with the 6 h treatment (Figure 3). The release of K^+^ on the surrounding environment states that the bacterial cell membranes were leaked upon treatment with endophytic secondary metabolites.

### 2.8. EA Fraction of APS1 Hampers the Central Carbohydrate Metabolism of the Pathogens

The EA fraction of APS1 directly hampers the central carbohydrate metabolism of the bacterial pathogens. The necessary enzymes, viz., PFK, ICDH and FBPase, were blocked by the action of endophytic metabolites. Figure 4 represents the inactivation of these necessary enzymes (Figure 4). The effect of these metabolites towards Gram-negative bacterial pathogens was very minimal, whereas the Gram-positive bacterial pathogens were highly affected. This type of findings suggest that the secondary metabolites produced by APS1 endophyte hampers the process of energy metabolism directly or indirectly by interfering the most vital carbohydrate metabolic pathways (33). At lethal concentrations, the organisms were unable to overcome the stress conditions and, as a result, the enzyme function reduced drastically.

### 2.9. Synergistic Action of EA Fraction of APS1 and Antibiotic Ciprofloxacin

In today’s modern world, in order to combat the day by day increasing parasitic infections of multi-drug resistant pathogenic microorganisms, it is quite necessary to use a mixture of bioactive compounds in order to minimize bacterial infection. In this paper, the synergistic action of APS1 and the popular antibiotic drug ciprofloxacin was tested by using them in variable combinations against MRSA. The results obtained from a checkerboard study revealed the fact that a combination of 7.81 µg/mL of the EA fraction of APS1 and 0.3 µg/mL of ciprofloxacin exhibits synergistic action against MRSA with ΣFIC of 0.42. The rest of the combinations did not exhibit any kind of antagonistic action (ΣFIC > 4). OD values on each row and column of the checkerboard reveal the growth of MRSA (Table 2).

### 2.10. OVAT Optimization

*Cochliobolus* sp. APS1 was treated in various fermentation conditions in order to check the maximum occurrence of antibacterial production in terms of the zone of inhibition and biomass development. Incubation time, temperature and medium pH were tested in a range from 3–11 days, 22–30 °C and 4–8, respectively. In each case, the supreme antibacterial and biomass production was noted at mid values, such as at 27 °C, 6.5 pH after 7 days of incubation. Below or above these levels, the performance dropped drastically (Table 3). Glucose, sucrose, rhamnose, maltose, and galactose were tested as additional sugar sources and yeast extract, urea, NH_4_NO_3_, beef extract, and peptone were tested as additional nitrogen sources. Finally, glucose and urea were selected (on the basis of the larger clear zone of inhibition) as additional carbon and nitrogen sources, respectively, and various concentrations (2–10 g% and 0.25–1 g%, respectively) of these nutrients were screened for their maximizing effect on antibacterial production. Optimum production was reported at 10 g% glucose and 0.8 g% urea concentrations. Other than these, salt concentrations on the medium significantly influenced performance parameters. NaCl at a concentration of 0.1 g% enhanced antibacterial production. After OVAT optimization, there was an increase of 21.06% (pre-optimized, 15.66 ± 0.33 mm clear zone of inhibition; optimized condition, 17.98 ± 0.58 mm clear zone of inhibition) antibacterial action in comparison to the un-optimized one.

Oxygen diffusion plays a key role on fungal growth and metabolism. Head space volume, medium depth and surface area of the medium influence oxygen availability on Erlenmeyer flask and these are directly proportional to medium volume. Different medium volumes were experimentally tested for finest activity and a 90 mL culture broth (230 mL head space volume) at a 250 mL (320 mL total flask volume) flask was found to be most suitable (Table 4).

### 2.11. Optimization by RSM

Coupled with OVAT, RSM was adopted using a three level Box–Behnken design involving four prime factors (glucose and urea concentration, fermentation time and medium pH) to elucidate the optimum antibacterial production from this organism at minimum effort. There was a sharp variation on response according to the different combinations of fermentation condition and five replicate of center points yielded the highest response (Appendix A). The predicted response Y for antibacterial production was reported in terms of coded factors as follows: Y_ZOI_ = −129 + 3.1 GC + 3.7 UC + 26.6 MpH + 3.5 FT − 3.1 GC*GC − 39.8 UC*UC − 2.09 MpH*MpH − 4.6 FT*FT − 8.3 GC*UC + 1.1 GC*MpH + 2.4 GC*FT+ −5.6 UC*MpH + 6.7 UC*FT + 2.50 MpH*FT
where Y_ZOI_ is the predicted zone of inhibition in mm (antibacterial action); X_1_, X_2_, X_3_, and X_4_ are coded factors of urea concentration (g%), glucose concentration (g%), pH of the fermentation medium and fermentation time (days), respectively. The goodness of fit of RSM and experimental outputs was conducted by regression analysis (Appendix A). The model F value (1476.13) was significant, having only 0.01% chance of error. The adjusted determinant coefficient (R^2^Adj) indicated a good correlation between the experimental and predicted values, with 99% chance of clarification of response by second order polynomial equation. Lack of fit F value (0.20) was not significant in a comparison to pure error and fitness of model was satisfactory. The model *p* value demonstrated its appropriateness to explain the response to antibacterial production. Lack of fit model *p* value (0.981 ˃ 0.05) reported good precision and uniformity of the investigated values. Here, the linear and two-way interactions between the four factors had a significant positive impact (*p* < 0.05) on antibacterial production except the interactions between medium pH–urea concentration and medium pH–fermentation time (*p* ˃ 0.05). Contour and 3D plots were created by Minitab to elucidate the interactions between the different variables for optimum antibacterial production (Figure 5A,B). There was an ignorable difference between the predicted (22.85 mm) and observed response (22.81 ± 0.16 mm) adopting the model’s fermentation stipulations, urea (0.74 g%) and glucose concentration (9.7 g%), medium pH (6.48), fermentation time (8.76 days).

### 2.12. Bio-Fim Inhibitory Activity of Cochliobolus sp. APS1

All the pathogens treated with the endophytic culture extract showed a significant reduction in biofilm formation (Appendix A). Gram-positive and Gram-negative pathogens formed a biofilm in the control situation in the absence of any endophytic extract. The antibiofilm effect was highest against *Bacillus subtilis* and *Shigella flexneri*, followed by *Escherichia coli*, VRSA, *Staphylococcus aureus*, *Proteus mirabilis*, and MRSA with an inhibition percentage ranging from 88.08 to 97.4, respectively. Biofilm inhibition was visualized by using crystal violet on 24-well plates (Figure 6). So, the antibacterial principles of *Cochliobolus* sp. Are efficient in biofilm inhibition.

### 2.13. Larvicidal Potency

The ethyl acetate extract of the endophytic *Cochliobolus* sp. APS1 exhibited antilarval potency against *Aedes aegypti* larvae of different stages (second instar stage, third instar stage and fourth instar stage) with an LC_50_ and LC_90_ values of 9.196 µg/mL and 15.970 µg/mL, 19.87 µg/mL and 34.16 µg/mL, 25.13 µg/mL, 48.09 µg/mL, respectively (Appendix A). X^2^ values were found to be insignificant at *p* ≤ 0.05 level. LC_50_ and LC_90_ values were calculated using the results obtained from percentage mortality. Larval death was reported after 24 h of treatment, whereas the control (EA + DMSO) showed no mortality. It can be observed from the results that the higher concentrations of the metabolite exhibited higher mortality values. The death of fungal pathogens started within 6 to 8 h of exposure of the ethyl acetate extracts of the endophytic fungi. More than 55–60% larval mortality happened within 11 to 13 h of treatment and zero survivors were found after the complete treatment of 24 h. So, the extracts act in a dose-dependent and time-dependent manner. 

Neurobehavioral toxicity was also detected among the isolates. After 30–45 min of treatment of the mosquito larvae with fungal ethyl acetate extract, abnormal restless movements were shown among the treated larvae. Extreme excitation frequencies, self-anal biting, regular irritations, coiling of the larvae body along with vibrating of the body parts (paralytic syndromes) were seen (Figure 7A). The fungal extracts showed inhibition of acetylcholine esterase enzyme action, which signifies the larvicidal potency of the extract. The fourth instar larvae of *Aedes aegypti* were used for this enzyme inhibition study.

The higher concentrations of ethyl acetate extract of APS1 cause the maximum inhibition of the acetylcholine esterase enzyme (Figure 7B). In the case of 100 µg/mL to 500 µg/mL extract application, the larval enzyme activity falls drastically (7.771 ± 0.011 µM/min/mg larval protein to 5.01 ± 0.003 µM/min/mg larval protein). Thus, a dose-dependent response was found in this case.

### 2.14. Identification of the Active Components

The Fourier transformed infrared spectrum of the *Cochliobolus* sp. MEAE revealed the presence of various absorption peaks. The major peak was observed at 3373.61 cm^−1^, which may be due to presence of O–H stretching present in case of aromatic compounds. The second moderate peak was at 2941.47 cm^−1^ representing the C–H stretching of alkane-like compounds. The peak at 1659.61 represents the N–H bending of the primary amines. Other necessary peaks with their type of vibrations were recorded in Table 5 and Appendix A.

The bioactive ethyl acetate fraction was tested for its metabolic constituents using GC-MS. In total, 25 compounds were identified by NIST library represented in Table 6. The majority of the components belong to long-chain hydrocarbons, mainly alkanes, such as tridecane, tetradecane, undecane, 10-methyl eicosane and aldehyde nonanol, as well as hexahydrofarnesyl acetone, i-propyl 12-methyl-tridecanoate, decyl decanoate, 2-cis-9-octadecenyloxyethanol, 1-ethylidene, 1-methylene-1H-indene, 2,4-dimethyl-3-hexanone being produced by the endophyte in culture broth. Tertiary alcohol 2, 6-dimethyl-7-octen-2-ol, also known as dihydromyrcenol (a monoterpenoid and tertiary alcohol), along with 3-methyl-1-butanol and 2-methyl-1-propanol were detected along with 2-benzothiophene having a similar odor to naphthalene. Dihydromyrcenol is a popular chemical used for surface sterilization, room freshener, as a fragrant in soap or perfumes, etc. [18]. Another naphthalene derivative (1,3-di-iso-propyl naphthalene) was found in minute amounts. The antifungal organic acid propanoic acid was detected along with phenol conjugates, such as p-cresol derivatives and phenol, 2-(6 bromoquinolin-8-yl). APS1 produced (2-aziridinylethyl) amine and phthalic acid in comparatively greater quantities (64.93% and 17.64%, respectively) than other secondary metabolites (Appendix A). The former one with an aziridine ring is mainly responsible for broad spectrum antibacterial activity of the isolate [19]. 

## 3. Discussion

Endophytes do not always tend to produce reproductive structures in vitro, thus making it difficult to categorize them into a particular taxon without rDNA analysis. The use of carnation leaf pieces agar (CLA) is an age-old technique to induce reproductive structure formation in sterile isolates. In this study, CLA was taken as the growth medium to identify the isolate APS1, but the endophytic fungi still did not produce any reproductive structure, thus making it difficult to identify by macroscopic and microscopic morphology, and finally rDNA techniques were utilized to identify the fungal taxon [20].

The endophytic *Cochliobolus* sp. APS1 possesses broad spectrum antibacterial activity against a variety of pathogenic microorganisms with an MBC value of 31.25–250 µg/mL. The occurrence of aziridine, 1-(2-aminoethyl)-, at the highest percentage as the prime one among the bioactive metabolites (relative abundance, 100%, and area percentage, 64.93) is responsible for the isolate’s antibacterial utility. Aziridine, a natural alkaloid (heterocyclic compound), possesses broad spectrum pharmacological activities, such as antimicrobial (against pathogenic microbes and cancer cell lines), antitumor, antileukemic, anthelmintic, and immunomodulatory, as well as being an enzymatic inhibitor, and it has been isolated from a wide group of terrestrial and aquatic organisms, including actinobacteria, mushrooms (*Agaricus silvaticus*), medicinal plants (*Allium cepa*, *Petasites japonicus* and *Nicotiana tabacum*) and sponge [13,21,22,23,24,25]. Its chemical analogues and derivatives are widely utilized clinically (as disinfectant and preservative) and commercially in disease control and in the fields of applied chemistry (for the synthesis of peptide derivatives and amino acid) based on their reactivity and unique pharmacodynamic action [13,25]. The well-known aziridine antibiotics, azirinomycin, ficellomycin and mitomycins-A, B, C, isolated from *Streptomyces aureus*, *S*. *ficellus* and *S*. *caespitosus*, are effective against a wide range of pathogenic microorganisms, such as *Staphylococcus aureus*, *Proteus vulgaris*, *Bacillus subtilis*, *Streptococcus faecalis*, *Klebsiella pneumoneae*, and *Aeromonas salmonicida*, respectively [13]. Beyond their direct use as antibiotics, they are utilized as disinfectants and blood preservers, in casein-like albuminous substances, dressings, floors, fruits, leathers, meat, medicinal instruments, portable water vessels, skins, textiles, wall, etc. [25]. Although aziridine ring compounds were isolated from a variety of natural origins, this is the first time aziridine was reported from an endophytic source. The endophytic *Cochliobolus* sp. APS1 produces long-chain carbon compounds and alcohols, which supports the concept of myco-diesel and is similar with the findings of VOCs (volatile organic compounds) produced by other potent endophytes *Phoma* sp., *Phomopsis* sp. and *Hypoxylon* sp., etc. Thus, it strongly correlates with the fuel potency of endophytes or the production of organic compounds (3-methyl-1-butanol, 2-methyl-1-propanol) that could be utilized in next-generation aircraft fuel as additives of gasoline [9]. Still a lot of experimental confirmation and fungal–plant environmental relations are needed prior to designating *Cochliobolus* sp. APS1 as an endophytic fuel-producing fungi. This is the first time to the best of our knowledge that aziridine has been obtained from any endophytic source with antibacterial, larvicidal and anti-biofilm potency.

Fermentation processes always need to be optimized in order to obtain the maximized product outcome in an economically feasible manner. It not only saves time, but also provides a crystal-clear idea about the cost–benefit ratio. In this study, the OVAT system coupled with RSM was also adopted for higher productions. There was a sharp increase in antibacterial action up to the 9th days of the fermentation process, but after that there was a decline in antibacterial production. Temperature positively influenced the secondary metabolite production, but up to a certain range (22–26 °C); beyond that, the antibacterial performance started to decline. The probable explanation could be that temperature plays a key role on the enzymatic function, either inducing or inhibiting its activity. Microbial fermentations are always manipulated by medium pH providing maximum productions at optimum situations or low yield at odd pH values, influencing enzymatic action and closing the metabolic channels [26]. Carbohydrates are the best preference for fungi to use as energy and carbon sources, leading to secondary metabolite synthesis [27]. Glucose, in terms of production parameter, ease of availability and cost effectiveness, was selected as the suitable supplementary sugar source and 10 g% was the best fitted ratio for fermentation purposes. Urea was used as an additional nitrogen source not just by providing nutrients for secondary metabolites production, but also acting as a necessary physiological property to cease contamination chances even in prolonged fermentation setups [28]. The rise in antibacterial activity upon treatment with NaCl was due to its increased membrane permeability, causing secondary metabolite excretions. Respiratory conditions are also needed to be at optimum situations to obtain the best antibacterial production. During fermentation conditions, to maintain the oxygen equilibrium, a gaseous exchange occurs between the medium surface and headspace gas [29]. In this case, a cotton plug acts as a mild barrier of free air passing from the external environment so the medium and head space volume corresponding to the surface area and medium depth regulate oxygen availability and influence fermentation conditions. An increase in medium depth and volume could negatively regulate oxygen diffusion and antibacterial production; finally, optimum conditions were achieved in a 50 mL fermentation medium below or above which production tapers down. OVAT optimization with RSM helps to construct the best suitable parameters for optimum fermentation yield [30,31]

The bioactivity of aziridine had been first reported almost five decades ago [24,32]. Antibiotic azirinomycin I and II (with an MIC value of 600–700 mcg/mL and 25 mm of clear zone of inhibition against *Pseudomonas aeruginosa* and *Proteus vulgaris*) were isolated from an actinomycete *Streptomyces aureus* MA-2941 that was cultured in a tomatopaste–oatmeal medium with an incubation period and temperature of 3–4 days and 28 °C, respectively [24]. Azirinomycin II was a 3-methyl-2(2H) carboxylic acid [33]. Another aziridine-based antibiotic azicemicin A (1) and B (2) obtained from *Amycolatopsis sulphurea* MJ126-NF4, with a medium requirement of yeast extract1%, glucose1%, galactose2%, dextrin2%, (NH_4_)_2_SO_4_0.2%, CaCO_3_0.2%, medium volume 110 mL in a 500 mL Erlenmeyer flask and the antibiotic was tested in an in vivo mice model with an amount of 150 mg/kg effective dosage against Gram-positive and mycobacterial pathogens [34]. *Streptomyces ficellus* is is known to produce aziridine antibiotic ficellomycin and expresses a wide range of antibacterial action against several fungal and bacterial pathogens(*Aspergillus* sp., *Hemophilus influenzae*, *E*. *coli*, etc.) and the highest inhibition was found at 0.05 mg/mL [34,35]. Madurastatin B_3_ from *Nocardiopsis* sp. LS150010 had anti-tuberculosis activity and its maximum antibiotic production was reported at 0.5% soluble starch, 2% dextrose, 1% soybean powder, 0.2% peptone, 0.2% yeast extract conc., 0.4% NaCl, 0.05% KH_2_PO_4_, 0.05% MSO4, 7H_2_O, 0.2% CaCO_3_, and pH7.8 [36].Choi et al. (2015) reported marinoazepinone and marinoaziridine from *Mooreiaalkaloidigena* and *M*. *catalinimonas* with broad spectrum antibacterial action [37]. Miraziridine A, a natural aziridinyl peptide (obtained from red sea sponge, *Theonella swinnoei*), is reported to be a potent protease inhibitor useful in the treatment of HIV [38]. Beyond these, 1-(4′-methoxyphenyl-aziridine) was obtained from *Abies webbiana* collected from the Sikkim Himalayan region in India [39]. In relation to the landmark discoveries in the field of aziridine, our present investigation in most of cases matches with the previous reports and seeks biotechnological interest. Endophytes are all-square in every aspect of modern biotechnology [40,41,42] and it is the first report of aziridine obtained from endophytic sources. 

Bacterial pathogens are tough to treat when they are engaged in a biofilm state. The MDR strains of *S*. *aureus*, *E*. *coli*, *P*. *mirabilis*, and *P*. *aeruginosa* are some of the major pathogens that take a huge toll in this respect in terms of infection in catheters, intubation components, pacemakers, etc. [43]. In this study, all the ten pathogenic microorganisms tested for biofilm formation were positive in the control set and are minimally to drastically inhibited upon treatment with the endophytic culture extract of *Cochliobolus* sp. APS1. The pretreatment by fungal metabolites is thought to be converting the abiotic surface unfavorable for cellular attachment, decreasing cell susceptibility towards surface adherence and promoting the detachment of bacterial cells [44]. Essential oils are widely used along with antibiotics to treat nosocomial infections related to biofilm forming microbes. Endophytes, as a new tool with a cocktail of novel compounds, may open up new frontiers in this respect.

Endophytes open up new fortunes for the medical world [45,46,47]. Our findings of the diverse activity exhibited by endophytes of *Andrographis paniculata* (Green Chiretta) are just a new addition to that fact. The findings suggest that the active constituent aziridine is responsible for broad spectrum antibacterial activity and its production enhances after the optimization of culture conditions that not only emphasizes its biotechnological acceptance, but also reduces cost issues. This is the first time that aziridine alkaloid (aziridine, 1-(2-aminoethyl)-) was reported to have been obtained from any endophytic source. In addition to antibiotic production, the isolate could solve issues related to biofilms on medical devices and diseases related to it. Its potency to check MRSA and VRSA widens its field of application. The extracts are potent larvicidal too. In other words, the endophytic *Cochliobolus* sp. APS1 is of biotechnological interest with its chemical constituents and bioactive potentialities. 

## 4. Materials and Methods

### 4.1. Isolation and Identification of Endophytic Fungi

*Andrographis paniculata* plants were collected from Tapobon forest of Paschim Medinipur district, West Bengal, India and brought to the Microbiology and Microbial Biotechnology Laboratory, Vidyasagar University, for further study. Small pieces of explants were surface sterilized by a series of surface disinfectant; first with running tap water; secondly with NaOCl (2–10%), thirdly with H_2_O_2_-hydrogen peroxide (3%), fourthly with 70% ethanol (for 5–10 s), and finally the tissue fingerprinting method was adopted and the explants were incubated on a water agar medium [48]. The water agar medium had previously been supplemented with antibiotic in order to avoid the isolation of bacterial endophytes. Aliquots with final wash liquid were incubated on PDA plates to confirm the non-epiphytic nature of the isolated endophyte. The isolation plates were placed on BOD incubator at 30 °C for 3–5 days. The fungal hyphae emerged from sterile explants were immediately transferred to a potato dextrose agar medium for pure culture and identification purposes. 

SEM studies were made for the structural confirmation of the isolate following the standard protocols. The samples were slowly dehydrated in different gradations of ethanol ranging from 10% to 90% to avoid the shrinking of the mycelial structures and then critically point dried, coated with gold and examined with a Zeiss EVO18 (Germany) scanning electron microscope following the standard protocols [49]. 

The isolate was devoid of any reproductive structures and was grown on a carnation leaf pieces agar medium in order to induce reproductive structure. rDNA based molecular identification was adopted for the identification of the sterile isolate [50].Partial sequencing of small and large subunit of ribosomal RNA gene along with complete sequence of ITS1, ITS2 and 5.8S ribosomal RNA gene were performed. Further analysis was conducted using the 617-base pair of consensus sequence. Sequences were submitted to GenBank. Sequences obtained in this study were compared to the GenBank database using BLAST. Thirteen sequences including APS1 were selected and aligned using multiple alignment software program Clustal W and the phylogenetic tree was constructed using MEGA 7 [51].

### 4.2. Study of Antibacterial Activity of the Cell Free Supernatant of Endophytic Fungal Isolate (Cochliobolus *sp.* APS1)

#### 4.2.1. Antibacterial Action of Endophyte *Cochliobolus* sp. APS1 

Fungal PDB blocks of adequate size (1 cm × 1 cm) were inoculated into potato dextrose broth (75 mL in 250 mL Erlenmeyer flasks) and kept in shaker incubator for 8 to 10 days with 80 rotation per minute and 27 ± 2 °C incubation temperature was maintained. The culture broth was centrifuged at 10,000 rpm for 20 min and again filtered using Whatman filter paper. The fungal biomass was expelled taking only the cell free culture extract. That aqueous extract was used for antibacterial test against bacterial pathogens, Gram-positive (*B*. *cereus* (ATCC 14579), *B*. *subtilis* (ATCC 11774), MRSA (ATCC 33591), *S*. *aureus* (ATCC 25923), and VRSA (clinical isolate)) and Gram-negative(*P*. *mirabilis* (ATCC 12453), *P*. *aeruginosa* (ATCC 9027), *V*. *parahaemolyticus* (ATCC 17802), *E*. *coli* (MTCC 4296), and *S*. *flexneri* (ATCC 12022)) pathogenic bacteria, by agar well diffusion technique [49]. A total of 50 µLof the cell free culture extract was added to the 5 mm diameter agar well in the nutrient agar plates previously inoculated with pathogenic bacterial strains and incubated at 28 °C for 24–48 h at optimum temperature (28 ± 2 °C). Ciprofloxacin, a standard antibiotic, was used as a positive control and only double distilled water was used as a negative control. Clear zones of inhibition (mm) were recorded.

#### 4.2.2. Determination of the Nature of Antibacterial Principles 

The antibacterial components of the endophytic fungal extracts were checked for their thermostability. The cell free supernatant was kept in boiling water bath for 10 min. Additionally, in order to check the proteinaceous nature of the antibacterial components, they were treated with proteinase K (1 mg/mL) for 2 h at 37 °C. The antibacterial activity of both heat-killed and proteinase-K-treated cell-free culture extract were studied against the pathogenic bacterium MRSA by agar well diffusion method with appropriate control [52].

#### 4.2.3. Extraction of Antibacterial Components Using Solvent-Ethyl Acetate

Ethyl acetate was used as a solvent for the extraction of antibacterial components in this study. Endophytic APS1 was grown on 2 L of potato dextrose broth for 14 days at 28 °C and a cell-free culture extract was obtained by filter-paper-based extraction followed by centrifugation at 10,000 rpm for 20 min. A total of 100 mL of the cell-free extract was mixed with 400 mL of EA (ethyl acetate) extract and shaken vigorously for 20 min in a clockwise and anti-clock wise pattern. The upper portion (ethyl acetate) of the liquid was collected using a separating funnel. A vacuum rotary evaporator was used in order to evaporate the ethyl acetate. The remnants were suspended in DMSO (Dimethyl sulphoxide) and stored for further bioactivity studies.

#### 4.2.4. Antibacterial Activity of EA Fraction of Endophytic APS1 

The DMSO dissolved EA extract of APS1 endophyte was subjected for antibacterial activity using different concentrations (0.005–10 mg/mL). Antibacterial activity was assayed using agar well diffusion method against all the Gram-positive and Gram-negative bacterial pathogens already mentioned. A total of 100 µL of pathogenic bacterial cultures were spread on the Petri plates poured with MH (Muller Hinton) agar medium. Wells were prepared using cork-borer and 50 µL of EA extract was added in each well. Only DMSO was used as negative control. After 96 h of incubation at 28 °C, clear zones of inhibition were observed and diameters were measured.

#### 4.2.5. Detection of MIC and MBC Values of Endophytic Metabolites 

Minimum inhibitory concentrations (MIC) value of EA fraction of APS1 against MRSA was calculated by counting the number of colonies forming units (CFU) after treatment. A total of 1% fresh bacterial culture (OD_620 nm_ = 0.5) was inoculated to 10 mL of MH broth taken in separate culture tubes [53]. Different concentrations (0.005–2 mg/mL) of EA extract of APS1 were used for the treatment of inoculated broth. Different concentrations (0.005–2 mg/mL) of the EA extract of APS1 were mixed with the inoculated bacterial cultures and were incubated overnight at 28 °C. Bacterial cultures treated only with DMSO was considered as control. The MBC values for each pathogenic bacterium were used to elucidate the bactericidal property of the antibacterial principles by marking the drastic change of CFU between the control and the treated one over time. The number of CFUs were counted after the spreading of bacterial cultures on the MH broth with the appropriate dilutions. The change in CFU values were recorded in a graph for clear understanding as a time killing kinetics [54]. The recorded MIC and MBC values were the lowest concentrations of the crude extracts that have bacteriostatic and bactericidal activity, respectively. The results were expressed in unit of µg/mL and presented as the mean of three independent experiments. 

#### 4.2.6. Time-Killing Kinetics 

In order to find out the bactericidal or bacteriostatic nature of the antibacterial principles on the human pathogenic microorganism, the EA fraction of APS1 was added to the mid-logarithmic phase (6 h after inoculation) of the bacterial culture. A total of 10 mL of the MH broth was inoculated with 1% bacterial culture (OD_620_ nm = 0.5) and incubated at 28 °C. After 6 h, the EA fraction of endophytic APS1 (dissolved in DMSO) was mixed with the treated set at MIC. The number of CFUs were counted at every two hours of interval by spreading the treated bacterial culture on MH broth after appropriate dilutions. A set comprised of only DMSO treatment was taken as control and another set with the standard antibiotic ciprofloxacin was taken as standard. Ciprofloxacin was used at different MICs against different bacterial pathogens. Finally, the bacteriostatic and bactericidal nature was determined by observing the growth curves of both the treated and control sets [54].

#### 4.2.7. Detection of Release of Intracellular Material of MRSA upon Treatment by *Cochliobolus* sp. APS1 EA Extract

The release of intracellular macromolecules, protein and nucleic acid were checked after treatment with the EA fraction of APS1. MRSA, VRSA, *P*. *mirabilis* and *Pseudomonas aeruginosa* bacteria were grown (OD_620 nm_ = 0.5) in 250 mL MH broth and bacterial cells were harvested by centrifuging the culture broth for 10 min at 6000 rpm. The cell pellets were taken and washed for two times with 50 mM Na-P buffer (pH 7.0). The cell pellets were resuspended in 1 mL of the same 50 mM Na-P buffer. Then, the pellets were treated with EA extract of APS1 at its MIC and MBC values. After 6 h and 24 h of incubation, the complete solution was centrifuged at 10,000 rpm for 10 min in order to obtain the cell free extracts. Protein and DNA concentrations in the mixture were determined following the methods of Lowry et al. (1951) and Burton (1965), respectively [55,56]. The set that was only treated with DMSO was considered as control. 

#### 4.2.8. Effect of APS1 EA Fraction on Bacterial key Enzymes

Four different Gram-positive and Gram-negative bacterial pathogens, MRSA, VRSA, *P*. *mirabilis*, and *Pseudomonas aeruginosa*, were treated with the EA fraction of APS1 (respectively, MIC and MBC) for 24 h. Cells were harvested by centrifugation at 6000 rpm for 10 min. Then, the harvested cells were washed with Na-P buffer solution (20 mM) and again suspended in one mL of the same buffer. After that, cells were ruptured (using sonicator) in order to obtain the cell-free extracts. Finally, supernatants were collected by centrifugation at 10,000 rpm for 10 min and used as crude enzyme. The most three physiologically valuable enzymes, FBPase (fructose-1,6-bisphosphatase), PFK (Phosphofructokinase), and ICDH (isocitrate dehydrogenase), were assayed following the methods of Mandal and Chakraborty (1993) [57]. Other necessary components, such as substrate, enzymes (cell-free extracts of individual bacterial pathogens) and co-factor were mixed in a cuvette. The rate of NADP reduction was followed at 340 nm in a UV-Vis spectrophotometer. The specific activity was calculated as nano moles of substrate consumed per min per mole protein and compared to the untreated control.

### 4.3. Determination of Synergistic Activity of EA Fraction of APS1 and Antibiotic Ciprofloxacin by Checkerboard Method 

The synergistic antibacterial activity of the *Cochliobolus* sp. APS1 EA fraction and broad spectrum antibacterial antibiotic ciprofloxacin against MRSA were tested following the checkerboard method proposed by Orhan et al. (2005) with slight changes [58]. The EA fraction of the endophytic fungi *Cochliobolus* sp. APS1 (0–1000 µg/mL) and the most effective antibiotic ciprofloxacin (0–100 µg/mL) were used at variable combinations. A total of 1 mL of MH broth was taken to 1.5 mL micro-centrifuge tube and then inoculated with 1% MRSA culture (OD_620 nm_ = 0.5). APS1 EA extract and ciprofloxacin were mixed at variable combinations. The mixture was incubated at 28 °C for 48 h. Cell pellets were drawn after centrifugation for 10 min at 6000 rpm and washed thoroughly two times in 50 mM Na-P buffer. Next, the bacterial cells were suspended in 1 mL of the same buffer and the OD values were measured at 620 nm in order to quantify the cellular amounts. In order to calculate the ∑FIC (fractional inhibitory concentration), different OD values were placed in the checker-board. ∑FIC was calculated by the following formula, −∑FIC= FIC of EA fraction + FIC of ciprofloxacin, where the FIC of the EA fraction or ciprofloxacin = MIC of the EA fraction or ciprofloxacin in combination/MIC of the EA fraction or ciprofloxacin alone. The combination of endophytic culture extract and ciprofloxacin was considered antagonistic when ∑FIC > 4, synergistic when ≤0.5 and indifferent when >0.5 [59].

### 4.4. Study of Inhibition of Bio-Film Formation

Biofilm formation of the human bacterial pathogens were tested using a 24-well polystyrene cell culture plate. A total of 1 mL of MH broth was poured to each well and then inoculated with 1% fresh culture of bacterial pathogens. The EA fraction of APS1 (dissolved in 10% sterilized DMSO) was added at different concentrations to different wells. After an incubation of 48 h at 27 °C, the broth was decanted from each and every well and then washed with sterilized distilled water without hampering the biofilm formation. After that, each well was dried and then washed gently with sterilized water without disturbing the biofilm. Next, the wells were stained with 1 mL of 0.1% crystal violet and kept in room temperature for 10 min. The crystal violet stain was washed, 1 mL of 33% acetic acid was added to each well and then kept for 30 min at room temperature providing mild agitation in order to extract the bound crystal violet from bacterial cells. The optical densities (OD) of acetic acid solution were then measured at 595 nm using UV-vis spectrophotometer. The inhibition of the biofilm was determined according to the formula [60] %Inhibition of biofilm formation= 100 − {(OD570 of sample/OD570 of control) ∗ 100}. The biofilm formation was classified at three levels: the highest one (OD570 ≥ 1), intermediate (0.1 ≤ OD570 < 1) and no formation of biofilm (OD570 < 0.1).

### 4.5. Detection of Antilarval Potency of Endophytic Isolates

#### 4.5.1. Collection of Mosquito Larvae and Evaluation of Larvicidal Potency

Mosquito larvae were obtained from agricultural fields of Fulpahari gram panchayat, Midnapore town, West Bengal, India and finally identified according to the key manuals of mosquito [61,62]. They were maintained in the Vidyasagar University, Department of Botany and Forestry, with a temperature of 26 ± 2 °C, a relative humidity of 77 ± 2%, and a photoperiod of 13:11 (Light/Dark). 

Next, the endophytic fungal mycelial extract was prepared in particular concentrations (100 µg/mL, 200 µg/mL, 300 µg/mL, 500 µg/mL) and the antilarval potency was tested against the second, third, and fourth instar larvae of *Aedes aegypti* (Dengue vector) according to the modified methods of [16]. The mortality rate along with number of larval survivors were recorded after the 24 h of continuous exposure of the ethyl acetate culture extract. The whole system was maintained at room temperature (25 ± 2 °C), covered with mosquito net and was totally kept away from sunlight. Experiments were performed in triplicate and the number of larval deaths were recorded (WHO, 2005). 

#### 4.5.2. Dose–Response Bioassay and Inhibition of Acetyl Choline Esterase Enzyme

The EA extracts of *Cochliobolus* sp. APS1 were dissolved in 10% DMSO and different concentration of the APS1 extracts were made in order to test the antilarval potency of the endophytic isolate. *Aedes aegypti* larvae were put in 250 mL of glass beakers filled with double distilled water and different concentrations (100–500 µg/mL) of metabolite extracts were prepared using 100 mL water [63]. The negative control of the experiment was DMSO distilled water. Each experiment was performed for three times. Abbott’s formula was adopted in order to calculate the mortality rate and survival rate after 24 h exposure of the APS1 extract to the larvae [64]. Larval test containers were kept at room temperature (26 ± 2 °C) for 24 h without any agitation. LC_50_ and LC_90_ values were determined using Probit analysis [65]. Percentage mortality was calculated using the following formula: Percentage mortality = (Number of dead larvae/Number of larvae introduced to the test) ×100.

### 4.6. Optimization for Antibacterial Production

The valuable growth parameters, such as incubation temperature, aeration, fermentation time, medium composition in terms of different types and concentration of carbon and nitrogen sources, and medium pH, were optimized by OVAT (one variable at a time) method [66]. OVAT coupled with RSM (response surface methodology) by Box–Behnken design was adopted for superior results following standard procedures [67]. The dry weight of mycelial biomass was also measured for this purpose.

### 4.7. Identification of the Active Components

#### 4.7.1. Fourier Transformed Infrared Spectroscopy (FTIR)

The dried powdered endophytic fungal EA fraction of *Cochliobolus* sp. APS1 was subjected to FTIR spectrometer analysis by scanning the active components at a range of 400 cm^−1^ to 4000 cm^−1^ at a resolution of 4 cm^−1^. KBr pellets were prepared (500 mg of sample was mixed with 300 mg of KBr and pelletized to 3 mm diameter) and measurements were performed on a Bruker Optics (Germany) [68].

#### 4.7.2. GC-MS (Gas Chromatography and Mass Spectrometry) of the Active Compounds

Fungal extracts (ethyl acetate) were subjected to column chromatography with varying ratios of polar and non-polar solvents [49] and the fractions were checked for their respective bioactivity. The most potent antibacterial fraction with anti-larval and anti-biofilm potentiality was checked for their active constituents by GC-MS analysis [4]. Semi purified crude extract was dissolved in 3 mL of methanol (GC grade, Hi-media) and analyzed by single quadrupole GC-MS system (USA, Waltham, Massachusetts, Thermo scientific). The instrument was configured with a DB-5 Ultra Inert column (30 m × 0.25 mm) for 22 min run of 1 μL sample (split-less flow) with an injector port and oven temperature of 240 °C and 50 °C, respectively, having 10 °C/min ramping time up to 260 °C with helium as the carrier gas. Mass fragmentation pattern was analyzed by X-Calibur software. The identification of the various compounds was based on the SI (similarity index) and RSI (reverse similarity index) value with the best matched compound in the NIST library.

### 4.8. Statistical Analysis

All experiments were performed in triplicate and the results are presented as means ± standard errors (SE). Data were analyzed by Prism GraphPad software version 9.2.0 (332) (San Diego, CA, USA). Minitab (version 20.2) statistical software was used for response surface methodology experiments (Box–Behnken design). Dose mortality regression was calculated using a log Probit model [65]. LC_50_ and LC_90_ values were calculated using the R-software for statistical computing [69] and codes within the package MASS containing the material are explained in [70].

## 5. Conclusions

Antimicrobial resistance and emergence of multi drug resistant microorganisms has become a serious issue in today’s world. The conventional strategies are becoming helpless day by day. So, it is necessary to switch to something new, novel, natural and evidently sustainable. The present study provides a new solution to this problem. Endophytic metabolites possess antibacterial, biofilm inhibitory and larvicidal potency that can be very useful in controlling the pathogenic micro-organisms and disease-causing insects. The bactericidal effect of the antibacterial principles towards the MRSA and VRSA pathogens is especially the key element of this present study. The antibacterial components were optimized and the antibacterial production was upscaled. This is the first time *Cochliobolus* sp. APS1 was isolated as an endophyte from *Andrographis paniculata* from the collection site (virgin forest patch of an unexplored region), the Tapobon forest region of West Bengal, situated at the eastern part of India. Additionally, the production and optimization of the aziridine alkaloid from an endophytic source was for the first time reported in this paper. In addition to bactericidal effects, the endophyte-derived compounds possess some biofilm inhibitory properties and larvicidal actions. So, the endophytic *Cochliobolus* sp. APS1 possesses immense bioactivity and produces multipotent bioactive metabolites, which may open up options for the sustainable development of compounds against the drug resistance phenomenon.

## Figures and Tables

**Figure 1 molecules-27-01459-f001:**
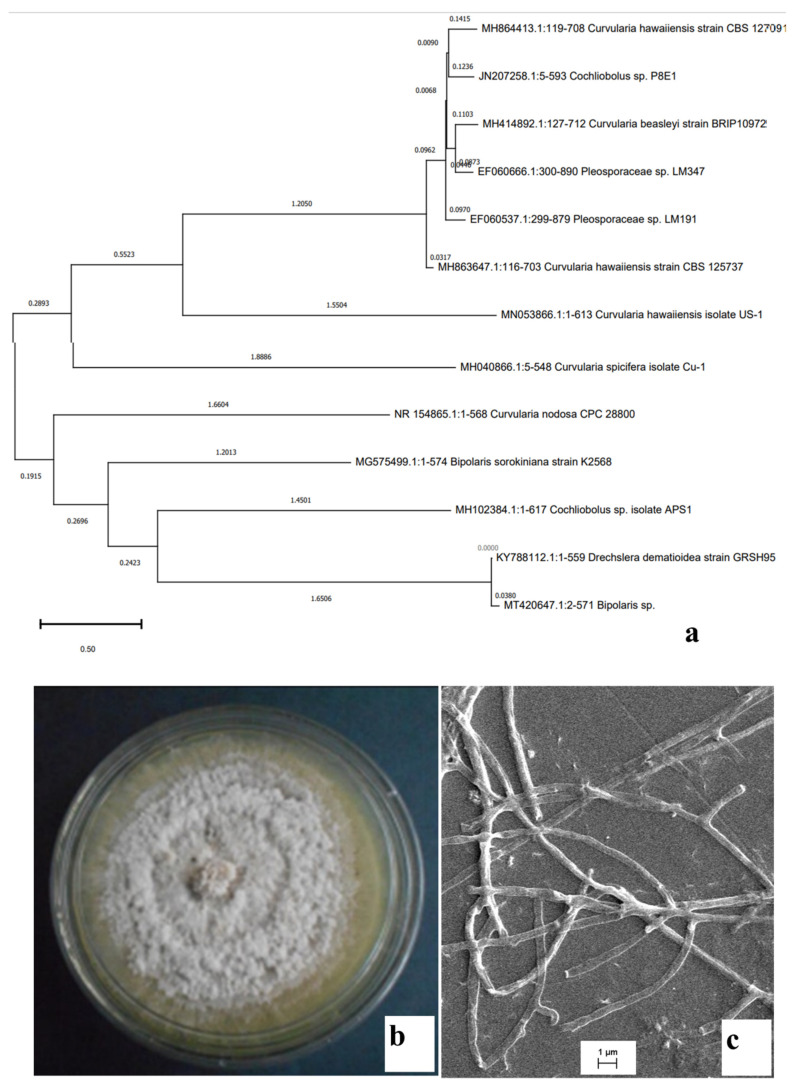
(**a**) Phylogenetic tree of the isolate *Cochliobolus* sp. APS1; (**b**) 8-day old colony of *Cochliobolus* sp. APS1 grown on PDA plates; (**c**) scanning electron micrograph of sterile hyphae.

**Figure 2 molecules-27-01459-f002:**
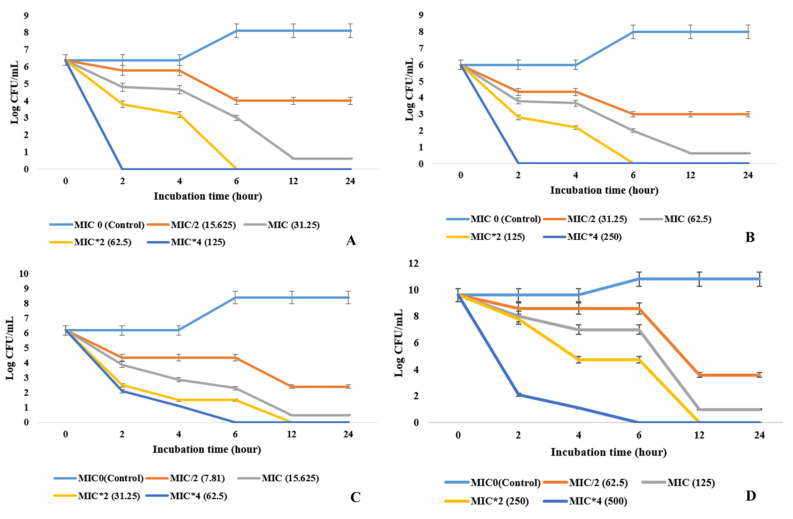
Killing kinetics of pathogenic microorganisms over time when treated with different concentrations of MIC values against pathogenic microorganisms: (**A**) *S. aureus*, (**B**) MRSA, (**C**) *B. cereus*, and (**D**) *E. coli.*.

**Figure 3 molecules-27-01459-f003:**
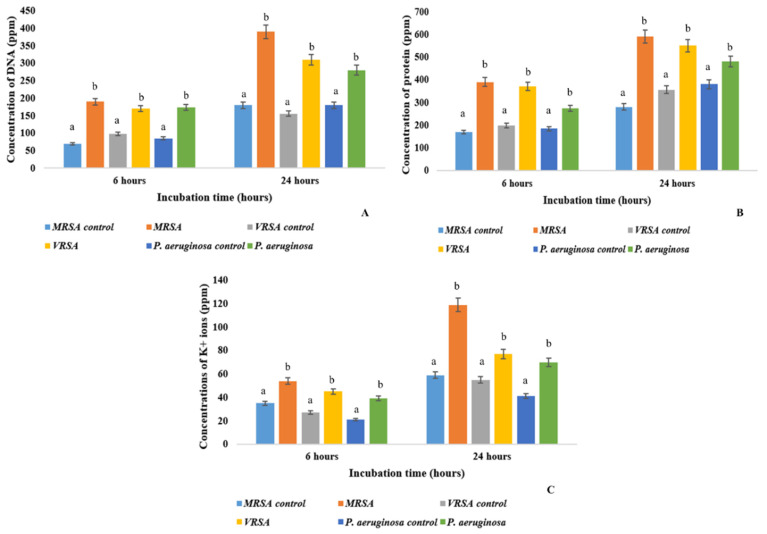
Leakage of intracellular macromolecules into the extracellular environment: (**A**) DNA content, (**B**) protein content, and (**C**) K^+^ ions. Values on the graphs are the means ± Standard error (SE) of the three replicates. Tukey’s multiple comparison test was performed. The different letters a, and b in each case (for each bacterial pathogen values at 6 h and 12 h) represents a significant difference between them (At, *p* < 0.05).

**Figure 4 molecules-27-01459-f004:**
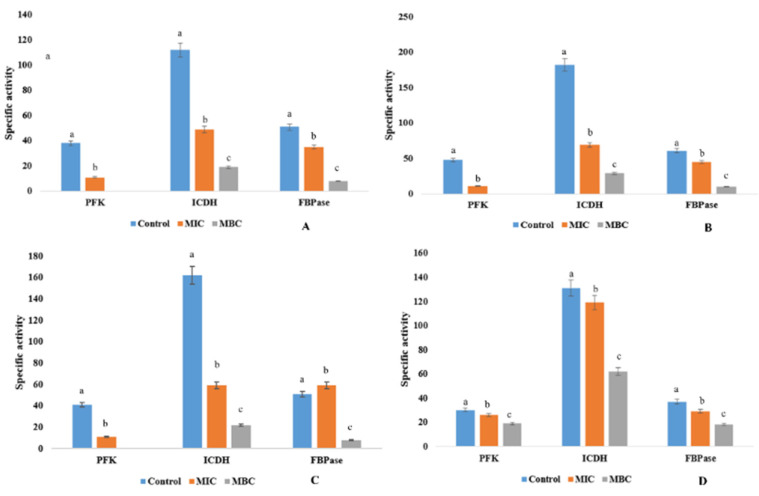
Change in the profile of central carbohydrate metabolic enzymes in pathogenic microorganisms upon treatment with ethyl acetate extract of APS1: (**A**) MRSA, (**B**) VRSA, (**C**) *P*. *mirabilis*, and (**D**) *P*. *aeruginosa*. Values on the graphs are the means ± Standard error (SE) of the three replicates. Tukey’s multiple comparison test was performed. The different letters a, b, and c in each case (Control, MIC and MBC values) represent a significant difference between them (At, *p* < 0.05).

**Figure 5 molecules-27-01459-f005:**
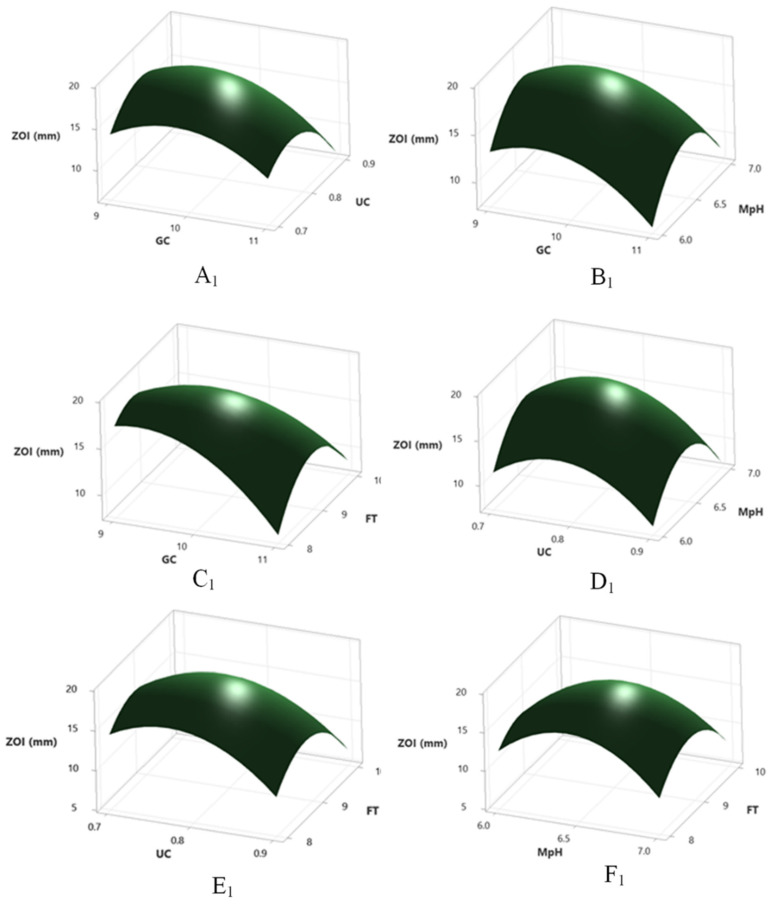
The 3D plot with 2D projection and contour plot showing the most important interactions of factors in RSM optimization of antibacterial activity by *Cochliobolus* sp. APS1. (**A1**,**A2**) between urea conc. vs. glucose conc. at fermentation time 9 days and medium pH 6.5; (**B1**,**B2**) between urea conc. vs. medium pH at fermentation time 9 days and glucose conc. 10; (**C1**,**C2**) between yeast extract conc. vs. fermentation time at glucose conc. 10 and medium pH 6.5; (**D1**,**D2**) between glucose conc. and medium pH at fermentation time 9 days and urea conc. of 0.8; (**E1**,**E2**) between glucose conc. vs. fermentation time at urea conc. 0.8 and medium pH 6.5; and (**F1**,**F2**) between medium pH vs. fermentation time at glucose conc 10 and urea conc. 0.8.

**Figure 6 molecules-27-01459-f006:**
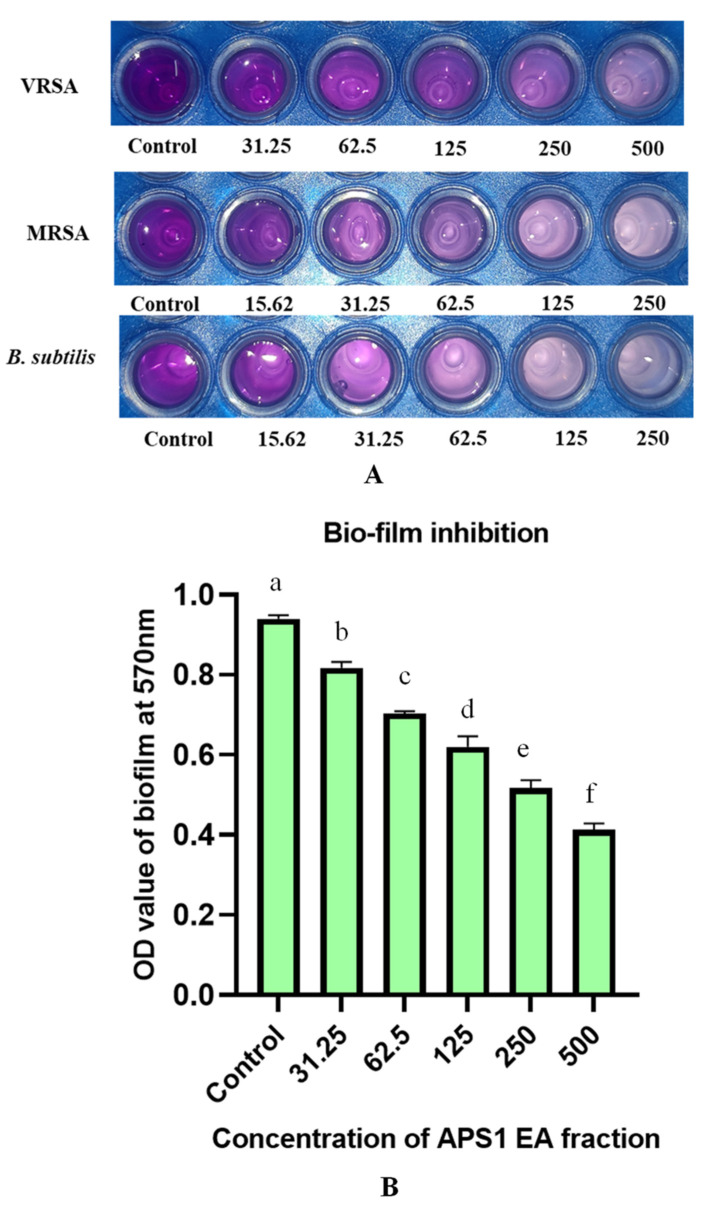
Biofilm inhibition by APS1 EA extract (**A**) against three pathogens (MRSA, VRSA, and *E*. *coli*) (crystal-violet-stained polystyrene plate showing bio-film formation as well as inhibition), (**B**) Change in OD values of the MRSA biofilm formation after treatment with different concentrations of the APS1 EA extract. Values on the graphs are the means ± Standard error (SE) of the three replicates. Tukey’s multiple comparison test was performed. The different letters a, b, c, d, e and f indicates significance difference in comparison to control (At, *p* < 0.05).

**Figure 7 molecules-27-01459-f007:**
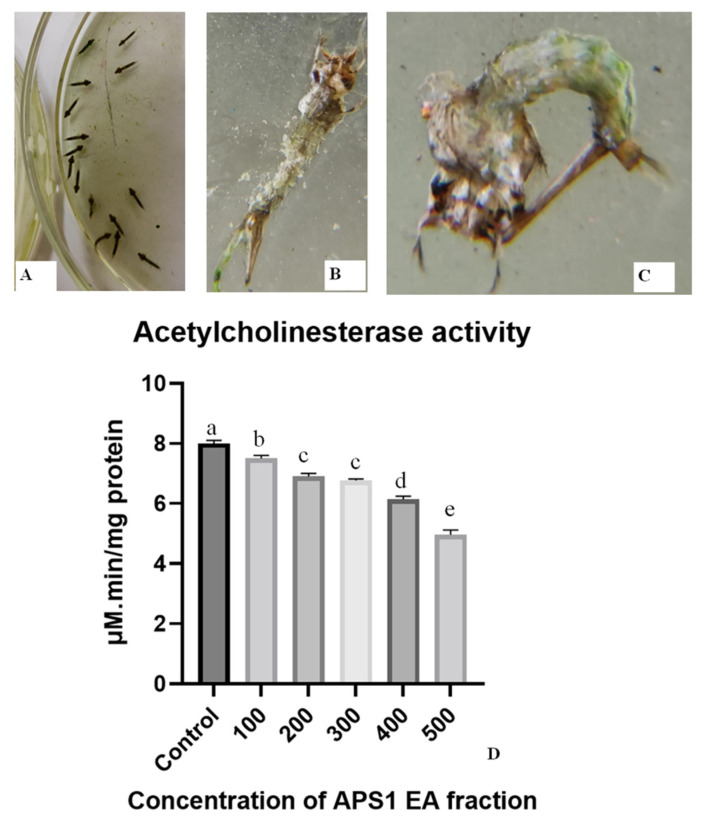
(**A**,**B**) Fourth instar larvae of *Aedes aegypti* after treatment with APS1 EA extract; (**C**) treated larvae showing degradation of body segments and (**D**) coiling of the larvae occurs; (**D**) Inhibition of the acetylcholinesterase (AchE) enzyme in the case of *Aedes aegypti* fourthinstar larvae after treatment with APS1 EA extract. Statistical values with same letter are not significantly different (according to Tukey’s HSD test at *p* < 0.05 (one way ANOVA)). Values on the graphs are the means ± Standard error (SE) of the three replicates. Tukey’s multiple comparison test was performed. The different letters a, b, c, d, e and f indicates significance difference in comparison to control (At, *p* < 0.05).

**Table 1 molecules-27-01459-t001:** MIC and MBC of the endophytic isolate (APS1) and standard broad-spectrum antibiotic against 9 pathogenic bacteria.

Sl. No.	Human Bacterial Pathogens	*Cochliobolus* sp. APS1		CPFX *	
		MIC Values (µg/mL)	MBC Values (µg/mL)	MIC Values (µg/mL)	MBC Values (µg/mL)
1	*B. cereus* (ATCC 14579)	15.62	31.25	0.7	0.7
2	*B. subtilis * (ATCC 11774)	15.62	31.25	0.8	0.8
3	*S.aureus* (ATCC 25923)	31.25	62.5	0.8	0.8
4	MRSA (ATCC 33591)	62.5	125	1	1
5	VRSA	62.5	125	1	1
6	*P. mirabilis * (ATCC 12453)	125	250	1.2	1.2
7	*P. aeruginosa* (ATCC 9027)	62.5	125	1.2	1.2
8	*E. coli* (MTCC 4296)	125	250	1.4	1.4
9	*S*. *flexneri* (ATCC 12022)	62.5	125	1.5	1.5

* CPFX: Ciprofloxacin.

**Table 2 molecules-27-01459-t002:** Checkerboard-based representation of the EA fraction of APS1 and ciprofloxacin (antibiotic) together on the growth of MRSA. In each combination, the growth is represented as OD values. (Different colours indicate the different OD values of bacterial culture (MRSA). Darker shades indicate higher OD values and a gradient of lighter shades indicates lower OD values.)

Ciprofloxacin (µg/mL)	**0.7**	0	0	0	0	0	0	0	0
	**0.6**	0.007	0	0	0	0	0	0	0
	**0.5**	0.098	0	0	0	0	0	0	0
	**0.4**	0.161	0.132	0.097	0	0	0	0	0
	**0.3**	0.209	0.168	0.101	0.051	0	0	0	0
	**0.2**	0.486	0.279	0.137	0.095	0.042	0	0	0
	**0.1**	0.792	0.478	0.371	0.279	0.097	0.037	0	0
	**0**	0.993	0.89	0.718	0.571	0.294	0.168	0.097	0
		**0**	**0.99**	**1.985**	**3.97**	**7.81**	**15.62**	**31.25**	**62.5**
	EA extract of APS1 (µg/mL)								

**Table 3 molecules-27-01459-t003:** Effect of different physical conditions and chemical supplements on biomass and antibacterial activity (ZOI, zone of inhibition) by *Cochliobolus* sp. APS1 against MRSA (methicillin-resistant *Staphylococcus aureus*).

Parameters	Effectors	Percentage Added (g%)	Biomass (g/L)	Antibacterial Activity (ZOI in mm)
Incubation time (in day)	3rd day	-	4.665 ± 0.076 a	4.33 ± 0.58 a
	5th day	-	5.565 ± 0.134 b	6 ± 0 b
	7th day	-	6.442 ± 0.064 c	9.67 ± 0.58 c
	9th day	-	6.015 ± 0.039 c	7.33 ± 0.58 d
	11th day	-	5.232 ± 0.076 b	7 ± 1 d
Incubation temperature (°C)	22	-	7.069 ± 0.045 a	9 ± 0 a
	24	-	7.855 ± 0.068 a	10.33 ± 0.58 b
	26	-	8.962 ± 0.049 b	13.33 ± 0.58 c
	28	-	6.062 ± 0.065 c	10.67 ± 0.58 b
	30	-	4.175 ± 0.034 d	8 ± 0 d
Initial medium pH	4	-	3.385 ± 0.024 a	3.33 ± 0.58 a
	5	-	5.005 ± 0.029 b	4.33 ± 0.58 b
	6	-	7.255 ± 0.068 c	7 ± 1 c
	6.5	-	8.995 ± 0.034 d	9 ± 0 d
	7	-	8.132 ± 0.068 d	6.33 ± 0.58 c
	7.5	-	7.995 ± 0.034 e	5.33 ± 0.58 e
	8	-	7.111 ± 0.03 c	5 ± 0 e
Additional carbon source	Starch	1	8.976 ± 0.02 a	9.67 ± 0.58 a
	Fructose	1	8.266 ± 0.036 b	8.67 ± 0.58 b
	Glucose	1	9.162 ± 0.073 a	11 ± 1 c
	Maltose	1	8.183 ± 0.037 b	9 ± 1 b
Additional nitrogen sources	Tryptone	0.3	9.172 ± 0.023 a	10.33 ± 0.58 a
	Urea	0.3	9.528 ± 0.029 a	13 ± 1 b
	NH_4_NO_3_	0.3	9.165 ± 0.024 a	9 ± 0.58 c
Glucose concentration	Glucose	2	9.522 ± 0.023 a	11.33 ± 0.58 a
		4	9.842 ± 0.024 a	12 ± 0 a
		6	9.965 ± 0.024 a	12.33 ± 0.58 b
		8	10.213 ± 0.021 b	13.33 ± 0.58 c
		10	10.841 ± 0.020 b	17 ± 0 d
		12	8.904 ± 0.023 c	11.33 ± 0.58 a
Urea concentration	0.25	-	9.212 ± 0.027 a	9.67 ± 0.58 a
	0.50	-	9.509 ± 0.029 a	12.33 ± 0.58 b
	0.70	-	10.164 ± 0.023 b	14.67 ± 0.58 c
	0.80	-	10.812 ± 0.030 b	16.67 ± 0.58 d
	1.00	-	8.304 ± 0.014 c	13.33 ± 0.58 e
Different metal ions	NaCl	0.05	10.134 ± 0.034 a	13 ± 1 a
	KCl	0.05	8.013 ± 0.009 b	9.67 ± 0.58 b
	MgCl_2_	0.05	6.106 ± 0.029 c	10.33 ± 0.58 c
	CaCl_2_	0.05	3.269 ± 0.027 d	11.33 ± 0.58 d
NaCl concentration		0.05	8.982 ± 0.003 a	14.33± 0.0.58 a
		0.1	10.009 ± 0.013 b	17.98 ± 0.58 b
		0.2	9.016 ± 0.040 a	15.67 ± 0.58 c
		0.3	8.090 ± 0.009 c	13 ± 1 d

One-way ANOVA (Tukey’s multiple comparison test) was performed to check the potential statistical differences in case of the biomass (g/L) and antibacterial activity (ZOI, mm) in different fermentation conditions (incubation time, temperature, conc. of sugar and nitrogen sources, etc.). There were valid statistical differences in most of the cases (*p* <0.05). The different letters a, b, c, d, and e indicate significant difference and the same letter at two positions indicate no statistical differences.

**Table 4 molecules-27-01459-t004:** **The** role of DO (dissolved oxygen) in the fermentation medium for antibacterial compound production.

Medium Volume (mL)	Total Volume of Flask (mL)	Head Space Volume (mL)	Medium Depth (cm)	Surface Area (cm)	Biomass (g/L)	Antibacterial Activity (ZOI, mm)
30	320	290	1.2	3.19	3.964 ± 0.123 a	6.6733 ±0.58 a
60	320	260	1.8	2.68	4.976 ± 0.017 b	8 ± 0.58 b
90	320	230	2.4	2.13	5.412 ± 0.170 b	9.33 ±0.58 c
120	320	200	3	1.8	5.009 ± 0.119 b	4.16 ±0.58 d

One-way ANOVA (Tukey’s multiple comparison test) was performed to check the potential statistical differences between the data (column wise) of biomass (g/L), and antibacterial activity (ZOI, mm) in different medium volume (mL). There were valid statistical differences in most of the cases (*p* < 0.05).The different letters a, b, c, and d indicate significant difference and the same letter at two positions indicate no statistical differences.

**Table 5 molecules-27-01459-t005:** Detection of functional groups present in endophytic fungal culture extract (*Cochliobolus* sp. APS1) using FT-IR spectra.

Wave Numbers (cm^−1^)	Peak Assignment	Mode of Vibration	Functional Group
3373.61	O–H stretching	Medium	Phenol
2941.47	C–H stretching	Medium	Alkenes
1659.61	C=C stretching	Medium	Alkane
1411.79	C–O bending	Medium	Inorganic carbonate
1261.36	C–O stretching	Medium	Alkyl aryl ether
1037.39	C–N stretching	Strong	Alkyl amine
911.75	O–H	Medium	Alcohol

**Table 6 molecules-27-01459-t006:** Bioactive compounds (secondary metabolites) produced by *Cochliobolus* sp. APS1 identified by GC-MS (NIST library).

Sl. No.	Name of the Compound	RT (min)	Area %	Ch. Formula	MW (g mol^−1^)
1	Phthalic acid	2.86	17.64	C_8_H_6_O_4_	166.14
2	2-((Z)-[(6-Bromo-8-quinolinyl)amino]methyl)phenol	3.34	9.15	C_16_H_11_BrN_2_O	326
3	3-methyl-1-butanol	4	0.64	C_5_H_12_O	88.14
4	1-ethylidene-1H-indene	11	0.64	C_11_H_10_	142
5	2-ethyl-p-cresol	12.40	0.64	C_9_H_12_O	136
6	2,6-dimethyl-7-octen-2-ol (Dihydromyrcenol)	12.74	6.49	C_10_H_20_O	156
7	2-ethyl-1-hexanol	13.98	064	C_8_H_18_O	130
8	2-mthyl-1-propanol	14	0.64	C_4_H_10_O	74.122
9	7-hexadecanal	14.1	0.64	C_16_H_30_O	238
10	2-cis-9-octadecenyloxyethanol	14.2	0.64	C_10_H_40_O_2_	312
11	Propanoic acid	15	0.64	C_3_H_6_O_2_	74.08
12	Methyl 2-ethylhexanoate	16.82	1.29	C_9_H_18_O_2_	158
13	Decyldecanoate	17.53	10.38	C_20_H_40_O_2_	312.5
14	(2-Aziridinylethyl)amine	18.31	64.93	C_4_H_10_N_2_	86
15	1,3-benzodioxan	21.28	1.94	C_8_H_8_O_2_	136
16	10 methyl eicosane	22.37	4.5	C_21_H_44_	296
17	1-methylene-1H-indene	26.77	0.64	C_10_H_8_	128
18	Undecane	30.02	1.29	C_11_H_24_	156
19	Nonanol	35.58	0.64	C_9_H_18_O	142
20	2-benzothiophene	36	0.64	C_8_H_6S_	134
21	2,4-dimethyl-3-hexanone	36.45	0.64	C_8_H_16_O	128
22	Tridecane	36.59	0.64	C_13_H_28_	184
23	Tetradecane	37	0.64	C_14_H_30_	198
24	Hexahydropseudoionone	37.44	0.64	C_13_H_26_O	198
25	1,3-di-iso-propylnapthalene	45.09	0.64	C_16_H_20_	212

## Data Availability

Not applicable.

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
