# Peer review of "Production of Bioactive Compounds with Broad Spectrum Bactericidal Action, Bio-Film Inhibition and Antilarval Potential by the Secondary Metabolites of the Endophytic Fungus Cochliobolus sp. APS1 Isolated from the Indian Medicinal Herb Andrographis paniculata"

_molecules, 2022, doi:10.3390/molecules27051459_

Round 1

Reviewer 1 Report

In this work, entitle “Production of bioactive compounds with broad spectrum bactericidal action, bio-film inhibition and antilarval potential by the secondary metabolites of an endophytic fungus Cochliobolus sp. APS1 isolated from an Indian medicinal herb Andrographis paniculata”, the authors report the identification of endophytes that display potent antibacterial, anti-biofilm formation, and antilarval activities. The authors identified Aziridine, 1-(2-aminoethyl) as the active compound responsible for the effects.

In general, the manuscript is very interesting and gives good information about endophytic production as a source of bioactive chemicals.

Some comments and suggestions for the manuscript are listed below.

Abstract

correct typing errors in the abstract

Introduction:

Line 48: define these abbreviations.

Line 69: “Mother Nature”… too informal and colloquial concept to be included in a scientific report

Line 77: Introduce why the study was performed in this plant species…

Line 80: what is APS1? Please define it

Results:

Line 99- 101: move this paragraph to the introduction section

Line 122: construction of phylogenetic tree... this is part of the materials and methods section

Figure 1a: Improve the resolution of the phylogenetic tree

Figure 2. include the statistical significance analysis

Tables 3 and 4: include the statistical significance analysis

Line 324: “Bo-film” - correct it

Materials and methods

Line 503: include the manufacturer information of the microscope used.

Author Response

Respected Reviewer 1, Thank you for your suggestions. We have made all the changes suggested by you.

Corrected the typing errors in the abstract.

In introduction the abbreviations have been elaborated as per your suggestion. (Line 48)

Line 69- the term “mother nature” has been changed according to your instruction.

Line 77- The study was conducted on the plant species as it has ethnomedicinal importance. The sentence that highlights the fact in the manuscript is “The plant Andrographis paniculata (Family-Acanthaceae) commonly called as “Green Chireta” was selected for isolation of endophytic fungi based on its immense medicinal importance in traditional and ethnic Indian Ayurvedic treatment as antibacterial and antioxidative agent [17].”

Line 80- Actually sir, APS1 is the strain name and number of the isolate (A-Andrographis, P-Paniculata S- Stem no. 1- the first isolate) Actually it is the short strain name of the isolate according to its source. Nomenclature have been done based on the source of the explant.

In results section Line no. 99-101- According to tour suggestion the portion has been moved to introduction.

Line 122 has also been displaced according to your valuable suggestion.

The resolutions of the figure have been improved to our best ability.

The statistical significance analysis has been given on the suggested graph.

In Table 3 and 4 also statistical significance have been allotted.

Line 324- Bio-film has been properly made.

The brand and country of origin of the microscope has been mentioned in the manuscript.

Reviewer 2 Report

Authors must have to explain production of ASP1 by novel endophyte and also have to compare with previous literature of all microbial and plant species.

Improve the quality figure 1 a

Improve the quality figure 2 and also add the bar in the figure

Remove the horizontal line in figures 3 and 4

Make table 1 in proper format

Try to send some tables in supplementary information

As I show many reference are old so please site the following latest publication to increase the quality of this manuscript

  1. Current advances of endophytes as a platform for production of anti-cancer drug camptothecin. Q Ruan, G Patel, J Wang, E Luo, W Zhou, E Sieniawska, X Hao, G Kai. Food and Chemical Toxicology 151, 112113
  1. Optimization of media and culture conditions for the production of tacrolimus by Streptomyces tsukubaensis in shake flask and fermenter level. G Patel, TP Khobragade, SR Avaghade, MD Patil, SH Nile, G Kai, .... Biocatalysis and Agricultural Biotechnology 29, 101803
  1. Production, immobilization and characterization of beta-glucosidase for application in cellulose degradation from a novel Aspergillus versicolor.C Huang, Y Feng, G Patel, X Xu, J Qian, Q Liu, G Kai. International Journal of Biological Macromolecules 177, 437-446

Author Response

Response to the respected reviewer 2

Thank you for your detailed supervision and fruitful suggestions.

The aziridine production by APS1 strain (Cochliobolus sp.) has been explained in the manuscript. Also, 5-6 examples of source, bioactivity and fermentation conditions of aziridine ring compounds have been incorporated in to the manuscript.

The figure (1a, 2) quality has been improved.

Error bard have been added in to the graph.

Horizontal lines have been removed from the graph.

Table 1 has been set to the appropriate manner.

Two table have been sent to supplementary portions.

The suggested latest publications have also been made.